# Systemic Inflammation, Endothelial Function, and Risk of Periodontitis in Overweight/Obese Adults

**DOI:** 10.3390/biomedicines11061507

**Published:** 2023-05-23

**Authors:** Oelisoa M. Andriankaja, Cynthia M. Pérez, Ashwin Modi, Erick L. Suaréz, Barbara A. Gower, Elaine Rodríguez, Kaumudi Joshipura

**Affiliations:** 1Center for Oral Health Research, College of Dentistry, University of Kentucky, Lexington, KY 40536, USA; 2Department of Biostatistics and Epidemiology, Graduate School of Public Health, Medical Sciences Campus, University of Puerto Rico, San Juan 00936-5067, Puerto Rico; cynthia.perez1@upr.edu (C.M.P.); erick.suarez@upr.edu (E.L.S.); 3Center for Clinical Research and Health Promotion, School of Dental Medicine, Medical Sciences Campus, University of Puerto Rico, San Juan 00936-5067, Puerto Rico; ashwin.modi@upr.edu (A.M.); elaine.rodriguez1@upr.edu (E.R.); kaumudi.joshipura@upr.edu (K.J.); 4Department of Nutrition Sciences, Division of Physiology & Metabolism, School of Medicine, The University of Alabama at Birmingham, Birmingham, AL 35294, USA; bgower@uab.edu; 5Department of Epidemiology, Harvard T. H. Chan School of Public Health, Boston, MA 02115, USA

**Keywords:** inflammation, endothelial, periodontitis, risk, obesity

## Abstract

The network interaction between systemic inflammatory mediators, endothelial cell adhesion function, and adiponectin as mediators of the association between metabolic diseases and periodontitis has not been evaluated. The objective of this study is to assess whether the interaction of baseline serum levels of TNF-α, hs-CRP, ICAM-1, VCAM-1, and adiponectin leads to periodontitis. Five hundred and ninety-seven overweight/obese (overweight: BMI 25 to <30 kg/m^2^; obese: >30 kg/m^2^) adults, aged 40–65 years, with complete 3-year follow-up data were included. Generalized structural equation models with negative binomial regression were used to estimate the regression coefficient (β) for the outcome number of teeth with probing pocket depth (PPD) ≥ 4 mm and bleeding on probing (BOP) at 3-year follow-up for a 1 standard deviation unit increase (Δ = +1SD) in each biomarker. After adjusting for multiple covariates, baseline ICAM-1 and VCAM-1 had significant direct effects on increased log-transformed number of teeth with PPD ≥ 4 mm and BOP (β: 0.16; 95% CI: 0.02–0.30; β: 0.15; 95% CI: 0.02–0.30, respectively). Baseline hs-CRP showed a significant indirect effect via ICAM-1 on the log-transformed number of teeth with PPD ≥ 4 mm and BOP (β: 4.84; 95% CI: 0.27–9.42). Thus, elevated serum ICAM-1 and VCAM-1 have a significant direct effect and increased hs-CRP has a significant indirect effect on the predicted level of periodontitis at the 3-year follow-up among overweight/obese Hispanic adults.

## 1. Introduction

Periodontitis is a polymicrobial chronic oral inflammatory disease that results in the destruction of the gingiva, periodontal ligament, and alveolar bone tissues supporting the tooth [1]. It is the major cause of tooth loss in adults and the most prevalent form of bone pathology in humans [2]. The prevalence of severe periodontitis in the dentate US adult population aged 30–90 years is around 8 to 10% [3]. Although bacterial pathogens are the major cause of periodontitis, their presence alone is not always sufficient to cause disease. The strength of the host’s immune response to the bacterial pathogens, which depends on a number of factors, such as genetic susceptibility, systemic health, and environmental factors (e.g., tobacco use and obesity), can contribute to the occurrence of disease and its severity [4]. The persistence of low-grade systemic inflammation, which can originate in part from adipose tissue-derived cytokines, is a risk factor for metabolic syndrome, type 2 diabetes, and CVD [5] and may also result in the development and progression of periodontitis [6].

Circulating cell adhesion molecules (CAMs), such as intercellular adhesion molecule (ICAM-1) and vascular cell adhesion molecule (VCAM-1), or adiponectin, an adipocyte-derived hormone, are involved in chronic inflammatory processes, which contribute to pathogenesis of cardiovascular disease (CVD), diabetes, and other diseases [7,8]. Periodontitis, a potential risk factor for CVD, has been suggested to be associated with serum levels of these molecules, adiponectin, and other inflammatory mediators, such as tumor necrosis factor-alpha (TNF-α) and C-reactive protein (CRP) as well [9,10]. Periodontal parameters, such as bleeding on probing and periodontal probing were found to be cross-sectionally associated with elevated serum soluble ICAM-1 (sICAM-1) and CRP, respectively [9]. Moreover, though the effect was within a limited time frame, periodontal therapy in healthy individuals or individuals with diabetes and periodontitis has shown to reduce systemic levels of these biomarkers [10,11]. These suggest a connection between chronic oral and systemic inflammation [12].

However, little is known about the opposite effects of regulation of possibly imbalanced chronic systemic inflammation in response to periodontal pathogens, which may lead to periodontitis, especially in overweight or obese status. The mechanisms of immune response involved in the pathogenesis of periodontitis are not limited to local mucosa only but also the systemic immune system, especially at a deeper periodontal tissue location [13]. Lappin and colleagues also observed a predominant recruitment of plasma cells in the gingival/granulation tissues of periodontitis lesions as compared to gingivitis lesions, and that VCAM-1 was recruited at the deep connective tissue, indicating the role of systemic immune response at that location [14]. A recent study suggested weight loss to increase serum adiponectin and reduce TNF-α. The elevation of adiponectin, in turn, was associated with reduction in periodontitis determined by probing depth and clinical attachment loss parameters [15].

Some notable studies have published the association of only one or a few types of biomarkers and periodontitis development [16,17,18,19,20,21], however, in reality, these mediators do not act separately but rather interact and function in concert, forming complex networks which may explain the pathogenesis of periodontitis [22]. Therefore, this study explores untouched complicated series of mediating pathways operating between these biomarkers leading to chronic periodontal inflammation or periodontitis. To explore underlying dispositions of biomarkers, it is hypothesized that serum inflammatory mediators and adiponectin might have direct or indirect effects via regulation of systemic endothelial cell adhesion molecule expression on periodontal tissue inflammation and destruction. Considering this, the objective of the present study is to assess whether the interaction of baseline serum levels of TNF-α, hs-CRP, ICAM-1, VCAM-1, and adiponectin leads to periodontitis among overweight/obese individuals.

## 2. Materials and Methods

### 2.1. Study Population

We analyzed data from a sample of overweight (BMI: 25 to <30 kg/m^2^) or obese (>30 kg/m^2^), non-institutionalized Puerto Ricans who participated in the three-year follow-up ‘San Juan Overweight Adults Longitudinal Study’ (SOALS) [23,24,25]. SOALS started in 2011 and included evaluations at baseline (2011) and after a three-year follow-up period starting in April 2014. The study was completed in 2016. SOALS participants’ inclusion and exclusion criteria are described in detail elsewhere [23,24,25]. SOALS participants met the following criteria at baseline: (1) age 40 to 65 years; (2) overweight or obese; (3) presence of at least 4 natural teeth and/or no braces or orthodontic appliances; (4) no diabetes or major cardiovascular disease. This study was approved by the human subject ethics board of the University of Puerto Rico, Medical Sciences Campus (UPR Institutional Review Board (IRB), approved on 7 February 2010, IRB #A4840109) and was conducted in accordance with the Helsinki Declaration of 1975, as revised in 2013. Informed consent was obtained from all participants prior to performance of the study procedures.

### 2.2. Biochemical Measures

All participants fasted at least 10 h prior to morning blood sample drawing using a standard protocol and silicone-coated sterile blood collection tubes (Becton Dickinson Vacutainer Systems, Franklin Lakes, NJ, USA). Blood samples were centrifuged at 3000 rpm for 15 min. The serum was frozen and stored at −80 °C. Baseline and follow-up serum samples were available from all participants.

Serum hs-CRP values were measured by a highly sensitive latex turbidimetric method by Beckman Coulter AU5421 K-assay (Beckman Coulter, Inc., 250 S. Kraemer Blvd., Brea, CA, USA) at the Immuno-Reference Laboratory in Puerto Rico. The intra- and inter-assay coefficients of variation (CVs) were 1.63% and 1.75%, respectively.

Serum levels of TNF-α, ICAM-1, and VCAM-1 were measured using the Meso Scale Discovery (MSD) multiplex method, an enzyme-linked immunosorbent assay (ELISA) that uses electrochemiluminescence as the signal to detect binding events [26]. The inter- and intra-assay CVs for each analyte were as follows: (1) TNF-α: 5.73% and 5.77%, respectively (minimum sensitivity was 1.30 pg/mL); (2) ICAM-1: 13.91% and 4.20%, respectively (minimum sensitivity was 1.72 ng/mL); and (3) VCAM-1: 10.80% and 3.43%, respectively (minimum sensitivity was 8.01 ng/mL).

Adiponectin was measured using radioimmunoassay with a Millipore human adiponectin RIA kit (Billerica, MA, USA). The inter- and intra-assay CVs were 9.07% and 6.04%, respectively, and the minimum sensitivity was 0.50 ug/mL.

### 2.3. Outcome Measurement

We assessed participants’ PPD and periodontal bleeding on probing (BOP) during the oral and dental examination (performed by one of three trained and calibrated dental examiners) at the baseline and follow-up visits. The detailed description of the NHANES III protocol used to measure PPD [27] is provided elsewhere [23]. Briefly, participants underwent a full-mouth dental examination during which their periodontal status was assessed at six sites per tooth for all-natural teeth (excluding the third molars), using a similar protocol at baseline and follow-up visits. We used a periodontal manual probe with 2 mm markings (Hu Friedy PCP2). The dark bands divided the probe at 2, 4, 6, 8, 10, and 12 mm, and the probe’s tip diameter was 0.50 mm. The periodontal probe was gently inserted to the base of the sulcus or pocket. The value of PPD, measured in millimeters, was rounded down if the reading fell between two probe markings [27].

We defined BOP based on the presence of bleeding at the buccal and lingual surface of each tooth when any site on those surfaces was bleeding after probing [24]. Ideally, we could have used any of the 6 tooth sites with BOP (interproximal distal at buccal side, mid-buccal, and interproximal mesial at buccal side, interproximal mesial at lingual side, mid-lingual, and interproximal distal at lingual side). However, due to high levels of bleeding at each site, the exact source of the blood flow was difficult to differentiate. Thus, we only recorded bleeding on any two of the six sites at the lingual and buccal surfaces of each tooth to increase measurement accuracy. Since we were interested in assessing the relationship with systemic inflammation during the time when periodontal tissue might be inflamed [28], which could represent the disease activity, periodontitis was classified based on a combination of the two parameters probing depth and BOP [9]. Periodontitis was evaluated at follow-up visits as the number of teeth with probing pocket depth (PPD) ≥ 4 mm and BOP [23].

The dental examiners and recorders were trained and calibrated by the NHANES reference examiner (Dr. Bruce Dye) before the study began. The dental examiners achieved a 96% agreement (within 1 mm of clinical attachment loss measure) with the reference examiner [29]. The intra- and inter-examiner variability for BOP measurement was unavailable, as the reproducibility of this parameter is very difficult to assess [30].

### 2.4. Other Covariates

Detailed information on collected covariates is provided in our recent work [23,24,25]. Information on the following covariates was gathered at the baseline and follow-up visits using an interview-administered questionnaire: socio-demographic characteristics (age, gender, marital status, and completed years of education) and lifestyle habits, such as smoking status, alcohol consumption status, and physical activity compliance according to the WHO guidelines (75 min of vigorous or 150 min of moderate physical activity per week) [31].

Data regarding history of medical conditions (e.g., hypercholesterolemia, hypertriglyceridemia, coronary artery disease, diabetes, and hypertension) and use of medications related to these health conditions (including lipid-lowering agents, anti-hypertensive drugs, and anti-inflammatory drugs) were also collected. Height and weight were measured twice to the nearest 0.1 cm and 0.5 kg, respectively, and the average of the two readings was used. Body mass index (BMI) was computed as weight in kilograms divided by height in meters squared (kg/m^2^). Waist circumference (WC) was measured twice, averaged, and categorized as high (women with a WC ≥ 88 cm or men with a WC ≥ 102 cm) and low otherwise [32].

Blood pressure was measured three times with 1–2-min intervals. Hypertension was defined as having a systolic blood pressure greater than or equal to 140 mm Hg, diastolic blood pressure greater than or equal to 90 mmHg, or being on anti-hypertensive medication use. Prehypertension was defined as having a systolic blood pressure of 120–139 mm Hg or diastolic blood pressure of 80–89 mm Hg in persons who were not receiving anti-hypertensive medications. Normotension was defined as having a systolic blood pressure below 120 mm Hg and a diastolic blood pressure below 80 mm Hg in the absence of anti-hypertensive medication [33].

Fasting serum glucose was measured using the Vitros System 250 instrument and insulin levels using a TOSOH analyzer. The CV of the glucose measurement within the laboratory was 1.7. The intra- and inter-assay CVs for insulin were 1.49% and 4.42%, respectively [23]. Baseline homeostatic model assessment (HOMA) of insulin resistance (IR) was computed as [fasting insulin (mU/L) × fasting glucose (mmol/L)/22.5] [34]. We determined fasting serum levels of triglycerides and high-density lipoprotein cholesterol (HDL-C) using spectrophotometry (Hitachi 704 Chemistry Analyzer, Roche Diagnostics, Indianapolis, IN, USA). The CVs within the laboratory for triglycerides and HDL-C were 1.5% and 3.2%, respectively. Serum level of low-density lipoprotein cholesterol (LDL-C) was estimated using Friedewald’s equation.

Baseline oral hygiene status was measured during the dental examination using the Silness and Loe plaque index (PI) [35], which is a visual assessment of the presence of dental plaque on six specific teeth [36]. Detailed information on the PI measurement is provided elsewhere [24]. The baseline number of sites with BOP or the number of teeth with PPD ≥ 4 mm and BOP were included as covariates in all analyses.

### 2.5. Model Building and Research Hypotheses

Appendix A summarizes possible associations between each systemic biomarker and periodontal disease according to the literature (Appendix A). We hypothesized that there are direct or indirect interaction effect(s) between serum inflammatory mediators, endothelial cell adhesion molecules, and adiponectin, which would lead to an increased risk of periodontitis among overweight/obese Hispanic adults at the three-year follow-up visit. More specifically, elevated baseline serum levels of TNF-α or hs-CRP, ICAM-1, and VCAM-1 and a low baseline serum level of adiponectin might have direct or indirect effects on periodontal tissue inflammation or destruction among overweight/obese Hispanic adults.

### 2.6. Statistical Analysis

Descriptive analyses were conducted. Binary indicator variables were dichotomized as below, at or above the median values of the outcome number of teeth with PPD ≥ 4 mm and BOP. Spearman’s rank correlation coefficients were computed between the biomarkers so that two highly correlated biomarkers would not be entered in the same model.

We used a generalized structural equation model for negative binominal regression [37] to assess the magnitude of all possible associations between the baseline systemic markers (i.e., all entered in the model) and the outcome at the three-year follow-up visit. This model is an appropriate choice given the overdispersed distribution of the expected values of the outcomes. Mediation analysis was used to explore different biological pathways possibly involved in the direct and indirect effects between the systemic biomarkers and the outcome [38]. The estimated regression coefficients indicate the risk of periodontitis development or progression at the three-year follow-up visit as a function of the systemic level of each baseline biomarker. Akaike’s information criterion (AIC) and the Bayesian information criterion (BIC) were used to compare the different models.

Potential baseline confounders were included in the model if they contributed at least 10% to the change in the estimated magnitude of the association between the baseline systemic biomarker of interest and the outcome at the follow-up visit [23]. Potential confounders included age, gender, smoking status, alcohol consumption status, educational level, oral hygiene status expressed by plaque index (PI), total number of natural teeth, physical activity, BMI or WC, serum levels of glucose, LDL-C, triglycerides, HDL-C, self-reported use of anti-inflammatory medications, hypertension, and HOMA-IR. The baseline number of teeth with PPD ≥ 4 mm and BOP was forced into each model [39]. STATA software version 13 was used for all statistical analyses [40].

## 3. Results

A total of 1206 participants completed the baseline examination in 2014 (Figure 1).

Of the total of 1206, 134 were excluded at baseline: 6 did not have serum samples, 122 had hemolyzed blood samples, and 6 had missing values for important key variables (e.g., mean attachment loss and smoking). Of the remaining 1072 participants, 446 were additionally excluded due to loss to follow-up (*n* = 245), lack of serum samples (*n* = 17), partly or fully hemolyzed blood samples (*n* = 139), missing values for serum glucose at 2 h (*n* = 11), or clinical attachment loss (*n* = 34). A total of 626 participants’ samples were sent to the University of Alabama at Birmingham for laboratory analysis. An additional 29 participants were excluded due to missing values for other key variables (e.g., physical activity level, alcohol consumption status, etc.). The remaining 597 participants were included in the analysis of TNF-α, ICAM-1, and VCAM-1. A total of 869 participants had serum hs-CRP assessed at the Immuno-Reference Laboratory in Puerto Rico.

The baseline characteristics of the study population are described in Table 1.

Participants with number of teeth with PPD ≥ 4 mm and BOP at or above the median (“high PPD group”) at the follow-up visit tended to be male, be former or current smokers, be less educated, drink more, exercise less, use fewer anti-inflammatory medications, and have higher BMI but lower WC compared to those with a number of teeth with the same conditions below the median value (“low PPD group”). Those in the high PPD group had higher baseline BP, HOMA-IR, and serum fasting glucose and triglycerides; they also had lower baseline HDL-C and adiponectin than those in the low PPD group. In addition, participants in the high PPD group had poorer baseline oral hygiene and a greater number of missing teeth, and they already had a higher baseline number of teeth with PPD ≥ 4 mm and BOP than those in the low PPD group.

Most biomarkers were higher (or lower for adiponectin) for those participants in the high PPD group than for their counterparts regardless of the time point (baseline or follow-up), except for serum VCAM-1 for participants in the high PPD group (Table 2).

All biomarkers, except hs-CRP, slightly or moderately increased from baseline to follow-up visit, regardless of the level of the outcome.

Table 3 displays Spearman’s rank correlation coefficients between the biomarkers and the outcome.

The coefficients ranged from −0.08 to 0.45 with the highest significant correlation between ICAM-1 and VCAM-1 (*r* = 0.45). Thus, two separate models were fitted to investigate the association between each of these adhesion molecules and the outcome.

Table 4a,b shows the generalized structure equation models for the effect of ICAM-1 and VCAM-1 on the outcome.

ICAM-1 showed a direct effect on the outcome, after adjusting for baseline age, gender, smoking status, alcohol consumption status, fasting glucose, education, physical activity, WC, anti-inflammatory agents, mean PI, baseline number of teeth with PPD ≥ 4 mm and BOP, and all the other biomarkers to keep them constant (Table 4a). More specifically, a 1 SD unit increase in ICAM-1 showed a significant increase in the log-transformed number of teeth with PPD ≥ 4 mm and BOP ($\beta_{\Delta =+1SD}:$ 0.16; 95% CI: 0.02, 0.30) at the follow-up visit. However, hs-CRP exerted an indirect effect via ICAM-1. ICAM-1 significantly mediates the relationship between hs-CRP and increase in the log-transformed number of teeth with PPD ≥ 4 mm and BOP ($\beta_{\Delta =+1SD}:\;$4.84; 95% CI: 0.27, 9.42). The mediating strong relationship is influenced by the significant relationship between hs-CRP and ICAM-1 ($\beta_{\Delta =+1SD}:\;$29.79; 95% CI: 17.66, 41.93) (Appendix A) and ICAM-1 and the increase in the logarithm (log) number of teeth with PPD ≥ 4 mm and BOP ($\beta_{\Delta =+1SD}:\;$0.16; 95% CI: 0.02, 0.30).

On the other hand, VCAM-1 showed a direct effect on the outcome (Table 4b, Appendix A). A 1 SD unit increase in VCAM-1 showed a significant increase in the log-transformed number of teeth with PPD ≥ 4 mm and BOP ($\beta_{\Delta =+1SD}:$ 0.15; 95% CI: 0.02, 0.30) at the follow-up visit (Appendix A). Nonetheless, the direct and indirect effects of other systemic inflammatory mediators and adiponectin were not significant. Additional adjustment of the models for factors such as baseline HOMA-IR, lipids (including triglycerides, LDL-C, and HDL-C), and number of missing teeth or hypertension did not significantly change the estimate of the association. Moreover, there is no significant change in association even by excluding participants with anti-inflammatory medicine.

## 4. Discussion

Our findings suggest direct significant associations between elevated baseline serum ICAM-1/VCAM-1 and an increased risk of periodontitis at the follow-up visit. Moreover, we found a significant indirect association between elevated baseline hs-CRP (via ICAM-1) and periodontitis development/progression at the follow-up visit. To the best of our knowledge, this is the first study to simultaneously assess the direct and indirect effects of systemic inflammatory mediators, adiponectin, or endothelial cell adhesion molecules on periodontitis development/progression. These results suggest significant direct effects of endothelial cell adhesion molecules on periodontitis development/progression. The finding of a direct effect of VCAM-1 on periodontitis corroborates that of a previous small cross-sectional study of aggressive periodontitis [41]. ICAM-1 and VCAM-1 gene polymorphisms have been observed in the Chinese adult population with periodontitis [42]. However, Schenkein et al. did not report an association between serum ICAM-1 and aggressive periodontitis in young adults recruited at the Virginia Commonwealth University School of Dentistry Clinics [41]. Glurish et al. failed to show associations between serum ICAM-1 or VCAM-1 and periodontitis in subjects aged 25 to 74 years residing in Erie County, New York and surrounding areas [43].

In contrast to our finding of a potential indirect effect of hs-CRP, a previous longitudinal study of 11,162 men in Nagoya, Japan indicated no association between baseline serum CRP and one-year development of periodontitis among men without periodontitis at baseline [44]. However, Meisel et al., who conducted a longitudinal study of tooth loss, suggested that obesity as a risk factor of tooth loss is particularly related to CRP in men but not in women [45].

Previous cross-sectional studies have also reported no association between TNF-α or adiponectin and periodontitis or aggressive periodontitis [46,47], similar to our null findings in this study. However, a recent case–control study indicated positive findings on TNF-α [48]. Zhu et al., in their recent systematic review and meta-analysis, found lower serum adiponectin in patients with periodontitis [49]. Our null findings may be attributed to our study population of overweight or obese individuals. Most previous studies with positive associations cited above consisted of lean, normal, or at most overweight individuals [49].

To our knowledge, few researchers have longitudinally assessed the association of systemic inflammation and endothelial function with periodontitis development/progression, and we expand this knowledge by demonstrating the association of serum hs-CRP, ICAM-1, and VCAM-1 with the progression of oral disease. Pink et al. recently reported potential long-term effects of systemic low-grade inflammation, as indicated by an increase in fibrinogen or WBC, on periodontal disease progression [50].

Unlike other periodontal parameters which could not distinguish past or present experience of periodontitis and periods of stability/activity, we evaluated periodontal inflammation by using a case definition that clinically indicates the activity or inflammatory status of the periodontium at the time we performed the baseline and follow-up dental examinations to determine probable disease progression. We did not assess the associations with clinical attachment loss (CAL), a parameter that shows periodontal tissue destruction, but may not represent the disease activity [28].

Our findings suggest a positive direction for the association between the direct/indirect effect of hs-CRP, ICAM-1, and VCAM-1 and periodontitis development/progression. Whether these associations operate in the opposite direction is unknown. However, our data showed that participants already had more periodontitis at baseline and the levels of the biomarkers at baseline could have been compromised due to being overweight/obese or due to their previous exposure to oral inflammation, thus representing a vicious cycle. Alternatively, they may have developed in parallel, meaning that, as the low-grade inflammatory state in overweight/obese individuals developed, so did the periodontitis. Consequently, separating the effect of one from another would not be feasible.

Moreover, the present study with respect to the direct/indirect associations between markers of systemic inflammation, endothelial function, and development/progression of periodontitis appears to support the existing evidence of an inter-relationship between metabolic disorders, CVD, and periodontitis [51], as these markers are compromised in both periodontitis and general medical conditions. Serum CRP-mediated inflammation can be inhibited by the anti-inflammatory action of lipid-lowering agents (LLAs), such as statins, in patients with atherosclerotic risk factors [52]. Interestingly, our findings from previous analyses of baseline data from the same study population also showed significant associations between self-reported LLA use and both reduced serum hs-CRP and periodontitis [24], suggesting again a possible simple pathway between CVD and periodontitis (i.e., without inferring any cause–effect association).

Thus, the imbalance in the inflammatory immune response could potentially explain, at least partially, the relationship between chronic metabolic diseases and periodontitis. In this case, periodontitis appears just to be another component (i.e., a symptom) of existing chronic metabolic disorders and could therefore just be a clinical local manifestation of systemic inflammation. In other words, individuals with inflammation are more likely to have periodontitis, as well as diabetes, metabolic syndrome, or CVD. Thus, the underlying chronic inflammation probably predisposes individuals to both local and systemic diseases.

A chronic inflammatory state in an overweight/obese condition may lead to an environment favorable to periodontal microbial growth and dysbiosis. In turn, persistent exposure to subgingival dysbiosis may lead to a pro-inflammatory endothelial cell phenotype associated with the loss of endothelial integrity, contributing to chronic periodontal inflammation and further tissue destruction. As a more elaborate mechanism, insulin resistance in overweight or obese individuals may interfere with endothelial cell (EC) function, specifically with the EC function’s critical effect on vascularization and, thereby, the inflammatory response to periodontopathogens [53]. In other words, upregulated inflammatory mediators, such as hs-CRP or the LPS, may stimulate the upregulation of the levels of VCAM-1 or ICAM-1, which favor leukocyte migration and inflammatory response to periodontal pathogens [54].

Findings from this study may have important public health impact(s) and value. Prevention or control of chronic metabolic disorders (e.g., metabolic syndrome, obesity, diabetes, and CVD), by prevention or control of systemic inflammation, may also prevent or control periodontitis, as systemic and oral diseases are, at least partially, linked by overall inflammation. In other words, the oral cavity is a mirror of the general health of the individual and the presence of periodontitis in the oral cavity may reflect existing or undetected chronic systemic inflammation, which is related to metabolic disorders. On the other hand, although not assessed in the present study, prevention and control of periodontitis may also improve systemic inflammatory disorders. Study limitations include restriction to overweight or obese Hispanic individuals, which might limit the generalizability of the findings. Nonetheless, similar associations, at least in some of these biomarkers, were observed in other populations [45]. Study strengths include the longitudinal study design, and to our knowledge, this is the first study using complex statistical models to understand the potential inter-relationships between systemic endothelial cell adhesion molecules and inflammatory mediators and development/progression of periodontitis.

## 5. Conclusions

Our findings stress the potential involvement of systemic inflammation and endothelial function in the pathogenesis of periodontitis. Baseline systemic inflammation and endothelial function were associated with an increased risk of periodontitis development/progression at follow-up in overweight/obese Hispanic adults. Further studies should be conducted to assess the potential effects of periodontitis on systemic inflammation to improve our understanding of the bidirectional link between oral and systemic inflammatory diseases.

## Figures and Tables

**Figure 1 biomedicines-11-01507-f001:**
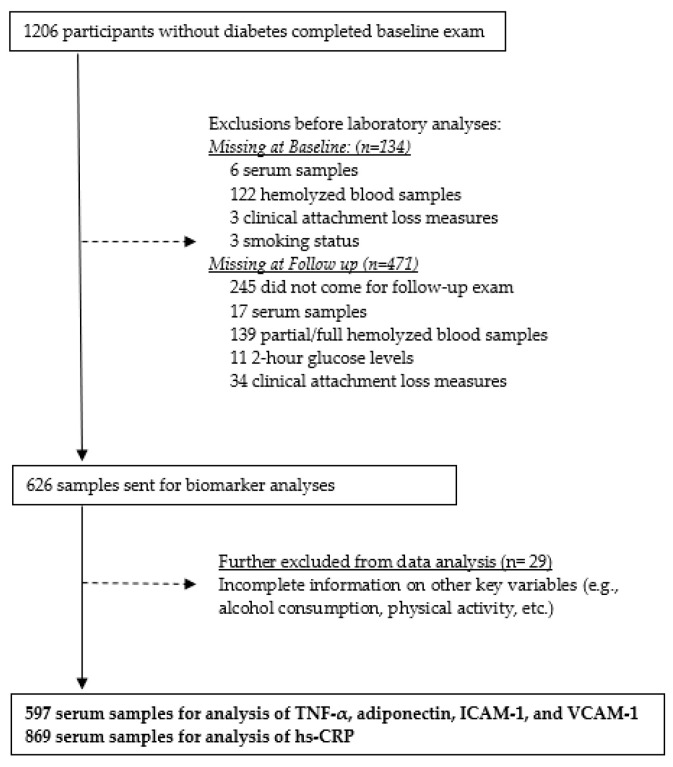
Flow Diagram of participants’ serum samples for biomarker analysis, San Juan Overweight Adults Longitudinal Study (SOALS), 2011–2016.

**Table 1 biomedicines-11-01507-t001:** Baseline characteristics of the study population by teeth with PPD ≥ 4 mm and BOP (high/low: < or ≥ median), * at follow-up visit (*n* = 597).

Teeth with PPD ≥ 4 mm and BOP at Follow-Up Visit		
Baseline Characteristics	<Median (*n* = 294)	≥Median * (*n* = 303)
	Mean ± SD or N (%)	
Age	50.6 ± 6.9	50.0 ± 6.6
Male gender	45 (15)	106 (35)
Smoking		
Never	210 (71)	191 (63)
Former	41 (14)	56 (18)
Current	43 (15)	56 (19)
Alcohol consumption		
Abstainer	135 (46)	131 (43)
Former	36 (12)	29 (10)
Current	123 (42)	143 (47)
Education (%)		
Less than high school	26 (9)	30 (10)
High school	62 (21)	88 (29)
More than high school	206 (70)	185 (61)
Physical activity	170 (58)	159 (53)
BMI (kg/m^2^)	32.4 ± 5.7	33.5 ± 6.3
High WC	258 (88)	251 (83)
High BP	130 (44)	146 (48)
Fasting glucose (mg/dL)	91.9 ± 8.7	93.7 ± 8.6
HOMA-IR	2.3 ± 1.5	2.7 ± 1.9
LDL-C (mg/dL)	122.8 ± 34.3	122.7 ± 31.8
Triglycerides (mg/dL)	141.5 ± 79.4	151.7 ± 78.7
HDL-C (mg/dL)	49.1 ± 13.0	47.5 ± 12.1
Use of anti-inflammatory agents	28 (10)	26 (9)
PI	0.7 ± 0.5	0.9 ± 0.7
Number of missing teeth	4.7 ± 4.4	4.4 ± 4.4
Number of sites with BOP	8.7 ± 8.8	16.7 ± 13.1
Number of teeth with PPD ≥ 4 mm and BOP	1.4 ± 2.8	5.4 ± 6.3

* The median of teeth with PPD ≥ 4 mm and BOP at follow-up visit was 1.

**Table 2 biomedicines-11-01507-t002:** Distribution of baseline and follow-up levels of the biomarkers by teeth with PPD ≥ 4 mm and BOP (high/low: < or ≥ median) at follow-up visit (*n* = 597).

Teeth with PPD ≥ 4 mm and BOP at Follow-Up Visit				
Biomarkers	<Median	≥Median	<Median	≥Median
	Baseline (Mean ± SD)		Follow-Up (Mean ± SD)	
hs-CRP * (μg/mL)	5.6 ± 6.7	5.5 ± 5.6	5.1 ± 6.5	5.2 ± 5.2
TNF-α (pg/mL)	2.5 ± 0.9	2.6 ± 2.4	2.7 ± 1.6	2.7 ± 2.2
Adiponectin (μg/mL)	9.6 ± 4.5	8.6 ± 4.8	10.2 ± 4.7	9.2 ± 5.3
ICAM-1 (ng/mL)	535.5 ± 147.2	550.7 ± 157.3	543.8 ± 152.6	572.5 ± 187.2
VCAM-1 (ng/mL)	601.9 ± 169.6	600.5 ± 163.6	630.6 ± 199.2	626.2 ± 190.2

* Serum hs-CRP level from *n* = 869.

**Table 3 biomedicines-11-01507-t003:** Spearman’s rank correlation coefficients among biomarkers (*n* = 597).

Baseline	TNF-α	hs-CRP	Adiponectin	ICAM-1	VCAM-1
TNF-α	-				
hs-CRP	0.08	-			
Adiponectin	−0.06	−0.08 *	-		
ICAM-1	0.29 **	0.31 **	−0.08	-	
VCAM-1	0.31 **	0.02	0.13 **	0.45 **	-
PPD ≥ 4 mm and BOP ^†^	0.08	0.01	−0.18 **	0.06	0.01

** p*-value < 0.05, ** *p*-value < 0.01; ^†^ PPD ≥ 4 mm and BOP: teeth with PPD ≥ 4 mm and BOP.

**Table 4 biomedicines-11-01507-t004:** Generalized structural equation models (GSEMs) of the interrelationship between the biomarkers (with 1 standard deviation increase), including ICAM-1 and vCAM-1, and the outcome number of teeth with PPD ≥ 4 mm and BOP (*n* = 597).

a. Models with ICAM-1 *				b. Models with VCAM-1 *			
**Direct Effect ^†^**							
**Teeth with PPD ≥ 4 mm and BOP**				**Teeth with PPD ≥ 4 mm and BOP**			
**Exposure**		**β ^‡^**	**95% CI**	**Exposure**		**β ^‡^**	**95% CI**
TNF-α		0.09	(−0.04, 0.21)	TNF-α		0.08	(−0.04, 0.19)
hs-CRP		−0.05	(−0.21, 0.12)	hs-CRP		−0.03	(−0.19, 0.12)
Adiponectin		−0.02	(−0.16, 0.13)	Adiponectin		−0.03	(−0.18, 0.12)
**ICAM-1**		**0.16**	**(0.02, 0.30)**	**VCAM-1**		**0.15**	**(0.02, 0.30)**
**Indirect Effect ^†^**							
**Exposure**	**Mediator**	**β ^‡^**	**95% CI**	**Exposure**	**Mediator**	**β ^‡^**	**95% CI**
	Adiponectin	0.00	(−0.09, 0.09)		Adiponectin	0.00	(−0.42, 0.43)
TNF-α	hs-CRP	0.00	(−0.02, 0.03)	TNF-α	hs-CRP	0.00	(−0.02, 0.02)
	ICAM-1	2.55	(−0.50, 5.60)		ICAM-1	3.98	(−0.17, 8.13)
	ICAM-1	−1.46	(−3.80, 0.82)		ICAM-1	2.62	(−0.48, 5.73)
Adiponectin	hs-CRP	0.03	(−0.08, 0.15)	Adiponectin	hs-CRP	0.02	(−0.09, 0.13)
	TNF-α	−0.00	(−0.11, 0.11)		TNF-α	−0.00	(−0.01, 0.01)
**hs-CRP**	**ICAM-1**	**4.84**	**(0.27, 9.42)**	hs-CRP	VCAM-1	1.04	(−1.23, 3.31)

* Each model adjusted for age; gender; smoking (never, former, current); education level (<12 years, ≥12 years); alcohol consumption (abstainer, former, current); anti-inflammatory medications (yes, no); waist circumference (>88 cm women, >102 cm men); fasting glucose level (mg/dL); physical activity (WHO definition: yes, no); mean plaque index; baseline number of teeth with PPD ≥ 4 mm and BOP; and all other corresponding biomarkers. ^†^ The direct and indirect effect on the association, for the mediation analysis, was assessed using GSEM for negative binomial regression. ^‡^ $\beta_{\Delta \;=+1SD}$: indirect association of exposure variable with outcome variable through given mediator after adjusting the covariates under GSEM-negative binomial regression model. The biomarkers with bolded estimates show statistically significant direct/indirect association with the outcome.

## Data Availability

The data presented in this study are available on request from the co-author, Kaumudi Joshipura (kaumudi.joshipura@upr.edu). The data are not publicly available due to privacy and ethical restrictions.

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
