# Peer review of "Systemic Inflammation, Endothelial Function, and Risk of Periodontitis in Overweight/Obese Adults"

_biomedicines, 2023, doi:10.3390/biomedicines11061507_

Round 1

Reviewer 1 Report

Periodontitis remains one of the most common oral diseases and one of the important role play inflammation. Moreover, inflammatory periodontal disease may be associated with an increased risk of cardiovascular disease. Studied inflammatory markers such as circulating cell adhesion molecules (CAM), intercellular adhesion molecule-1 (ICAM-1), or vascular cell adhesion molecule-1 (VCAM-1), are related to obesity and have been suggested as potential candidate markers of endothelial dysfunction, which contribute to the pathogenesis of cardiovascular diseases. In the presented manuscript serum levels of VCAM-1, ICAM-1 and hsCRP have been recognized as predictor markers of periodontitis.

The manuscript is well written, with five paragraphs describing particular elements of the work and relevant reference part.

One issue could be addressed more in the discussion: What molecular mechanisms could be responsible for the results obtained by the authors?

Reviewer 2 Report

Abstract - define the objective: To study the interaction of baseline serum levels 24 of TNF-α, hs-CRP, ICAM-1, VCAM-1, and adiponectin leads to periodontitis. Define overweight/obese defination. 

Introduction - Put referencves to justified that previous studies have evaluated the association between only one biomarker at a time and periodontitis development. Spell out the research gap.  The last paragraph should move to discussion, and replace with a statement on the objective of this study.

Method - Do not mix up method and results. You do not know the results (no. of participants  should be in results after you did the study). and Improve the quality of Figure 1. 

Results - Table Font size is small and quality is poor. Combine and summarizes the many tables.

Discussion - Please discuss the values and impact of this study.

Conclusion - Tpoo long and should be address the aoim only.

Some references are old, such as the one in 1985.
